# Sedation with Propofol plus Paracetamol in External Cephalic Version: An Observational Study

**DOI:** 10.3390/jcm11030489

**Published:** 2022-01-19

**Authors:** Javier Sánchez-Romero, Jesús López-Pérez, Ana Belén Flores-Muñoz, María Josefa Méndez-Martínez, Fernando Araico-Rodríguez, Jaime Mendiola-Olivares, José Eliseo Blanco-Carnero, Luis Falcón-Araña, Aníbal Nieto-Díaz, María Luisa Sánchez-Ferrer

**Affiliations:** 1Department of Obstetrics and Gynecology, ‘Virgen de la Arrixaca’ University Hospital, 30120 Murcia, Spain; araicos33@hotmail.com (F.A.-R.); eliblanco@um.es (J.E.B.-C.); anibal.nieto@um.es (A.N.-D.); marisasanchez@um.es (M.L.S.-F.); 2Department of Obstetrics and Gynecology, The University of Murcia, 30011 Murcia, Spain; 3Department of Anesthesiology, ‘Virgen de la Arrixaca’ University Hospital, 30120 Murcia, Spain; jesuslpmed@gmail.com (J.L.-P.); anafloresmuz@gmail.com (A.B.F.-M.); mariajose.mendez11@gmail.com (M.J.M.-M.); falconable@gmail.com (L.F.-A.); 4Division of Preventive Medicine and Public Health, School of Medicine and Biomedical Research, Institute of Murcia (IMIB-Arrixaca-UMU), University of Murcia, 30120 Murcia, Spain; jaime.mendiola@um.es

**Keywords:** propofol, ECV, sedation, breech

## Abstract

Although the influence of neuraxial anesthesia or sedation with remifentanil in external cephalic version (ECV) is widely known, ECV results using propofol have not been previously analyzed. This study aimed to evaluate ECV outcomes when propofol was used. An observational analysis of ECV was performed between 1 January 2018 and 31 December 2020. ECV was accomplished with tocolysis and propofol. One hundred and thirty-one pregnant women were recruited. The propofol mean dose was 156.1 mg (SD 6.1). A cephalic presentation was achieved in 61.1% (80/131) of the pregnant women. In total, 56.7% (38/67) of pregnant women with cephalic presentation at labor had a spontaneous delivery, 26.9% (18/67) had an operative delivery, and an intrapartum urgent cesarean section was performed in 16.4% (11/67). In total, 46 pregnant women (35.9%) were scheduled for an elective cesarean section due to non-cephalic presentation. The emergency cesarean section rate during the following 24 h was 10.7% (14/131). A major ECV complication arose in 15 cases (11.5%). ECV outcomes when propofol was used seems to be similar to those with other anesthetic adjunct, so sedation with propofol could be an adequate option for ECV. More studies are needed to compare its effectiveness with neuraxial techniques.

## 1. Introduction

Breech presentation affects 3–4% of singleton term pregnancies [1,2]. External cephalic version (ECV) is a procedure for modifying the fetal position and achieving a cephalic presentation. The objective of the ECV is to offer an opportunity for cephalic delivery to occur which, as widely known, is safer than breech vaginal delivery or cesarean section [1]. The use of an external cephalic version in breech presentation, according to the WHO [2], certainly reduces the incidence of cesarean section, which is of special interest in those units where vaginal breech delivery is not a common practice.

ECV is usually performed before the active labor period begins. Factors associated with a higher ECV success rate include the following [3,4,5,6]: multiparity, a transverse presentation, black race, posterior placenta, and amniotic fluid index higher than 10 cm.

Certain interventions may help in ECV [4,6,7] such as tocolysis or analgesia. Ritodrine not only is considered a safe tocolytic agent but also improves the ECV success rate [8]. Other tocolytic agents studied in ECV are nifedipine [8], atosiban [7], nitroglycerine [9], or other β-agonists [9].

Regarding analgesia in ECV, some interventions have been analyzed such as sedation or neuraxial anesthesia. The use of anesthetic adjunct for the procedure, as systemic opioids or neuraxial anesthesia increases the rate of successful ECV by as much as 60% [10,11,12,13]. When neuraxial anesthesia and intravenous remifentanil in ECV were compared, no differences were found [10].

Propofol is a widely used agent in obstetric anesthetic management [14]. It has many advantages over benzodiazepines and/or opioids, including rapid onset of action, short half-life, and fast recovery [14]. To the best of our knowledge, ECV results using sedation with propofol have not been previously analyzed, even though propofol is a common agent used for non-obstetric cases.

The main objective of this study was to analyze ECV outcomes when propofol was used as anesthetic adjunct for the procedure. As a secondary objective, we described the delivery mode when propofol was used in ECV. We hypothesize that propofol must be a safe agent for ECV that may reach an adequate maternal unconscious state with similar results to other analgesic procedures published.

## 2. Materials and Methods

This is an observational study of ECV performed in ‘Virgen de la Arrixaca’ University Clinical Hospital in Murcia (Spain) between 1 January 2018 and 31 December 2020 [15]. The ECV and obstetric outcomes of the patients in whom propofol was used were analyzed. This center is the largest maternity department in Spain, with approximately 7000 births per year. This manuscript adheres to the applicable STROBE guidelines. This study was approved 30 April 2020 by the Clinical Research Committee of the ‘Virgen de la Arrixaca’ University Clinical Hospital (2020-5-6-HCUVA). Written informed consent was obtained from all participants.

The ECV procedure was performed by the members of an ECV team. Each procedure was performed by two obstetricians of the Maternal-Fetal Unit in the obstetric operating room with the presence of an anesthesiologist and a midwife. Obstetricians, anesthesiologists, and midwives who are members of the ECV team were super-specialized professionals and they had more than seven years of experience in ECV. Women were recruited during the third-trimester obstetric evaluation at 36 weeks gestation.

ECV was offered to every pregnant woman with non-cephalic presentation and no absolute contraindication for vaginal delivery. Women were excluded in cases of severe pre-eclampsia, recent vaginal bleeding, confirmed rupture of membranes, and when an absolute indication for cesarean section was identified (i.e., placenta previa).

In the consult, all pregnant women were asked about personal and obstetric history. An ultrasound assessment for studying the fetal position, fetal biometry, amniotic fluid, and placental position was performed in the consult.

If the woman was eligible and informed consent was obtained, ECV was performed at 37 weeks gestation. All women were asked to fast for eight hours before the procedure. Before ECV was performed, the anesthesiologist evaluated the pregnant women. The women were asked to empty their bladders. Just before the procedure, 0.2 mg/min of ritodrine was intravenously administered for 30 min [15].

In the operating room, maternal vital signs were monitored (heart rate, EKG, temperature, noninvasive blood pressure, and oxygen saturation). The woman was positioned in Trendelenburg (15°). Paracetamol 1 g was intravenously administered as an analgesic agent. Propofol was used as a sedative agent. During the procedure, there was continuous feedback between obstetric and anesthesiologist teams.

With the objective of achieving an unconsciousness state, an initial dose of 50 mg of propofol was administered as a bolus at the beginning of the procedure. Sedation was maintained with additional boluses of 10 to 20 mg propofol were administered at intervals of at least 20 to 30 s, as needed.

Two ECV attempts following the forward roll technique were performed by two experienced obstetricians. Immediately after the procedure, fetal wellbeing was assessed with a continuous cardiotocograph register during the following four hours. Anti-D was given to rhesus-negative women. Then, 24 h after the procedure, fetal well-being was reassessed with continuous monitoring for 1 h. If any complication occurred immediately after the procedure, an urgent cesarean section was performed.

ECV is considered successful when a cephalic presentation is achieved. Clinically relevant hypotension was considered if systolic blood pressure (SBP) was below 90 mmHg or the fall of at least 20% of SBP [16].

Maternal hypotension is a common adverse consequence of both neuraxial analgesia and sedation. Maternal hypotension may be associated with non-reassuring fetal heart rate (FHR) pattern, maternal discomfort, nausea, and vomiting.

A major complication after ECV was considered when any non-reassuring FHR pattern (i.e., bradycardia longer than 8 min), major vaginal bleeding, maternal instability, or any critical emergency arose during the 24 h after ECV.

Clinical data were recorded prospectively on all referrals. Anesthesia data were recorded retrospectively from the anesthetic register. Data on pregnancy outcomes were collected from hospital obstetric and neonatal records. Continuous variables were assessed for normality with the Shapiro–Wilk test. Data analysis was performed using SPSS version 25.0 (SPSS Inc., Chicago, IL, USA) and R version 3.6.2 (https://www.r-project.org/, accessed on 4 February 2021).

## 3. Results

Although 242 pregnant women underwent ECV during this period, propofol was used in 131 cases. The characteristics of pregnant women who underwent ECV with propofol are shown in Table 1. The mean age was 32.1 years (SD 0.4). In total, 68.7% of the women were nulliparous. Only four pregnant women (3.1%) had a previous cesarean section. The mean estimated fetal weight at the third trimester was 2754.4 g (SD 29). The mean maternal BMI was 27.8 Kg/m^2^ (SD 0.4).

ECV was performed at 37.3 weeks gestation (SD 0.04) on average. The mean propofol dose was 156.1 mg (SD 6.1). Premedication drugs (atropine, midazolam, or fentanyl) were used in 48 cases (36.9). Although 24 pregnant women (18.3%) suffered hypotension during the procedure, just one required vasoactive drugs (0.8%). Thirteen pregnant women (9.9%) experienced nausea and vomiting and they were administered antiemetic drugs (ranitidine and metoclopramide).

ECV was successful in 80 pregnant women (61.1%). In nulliparous women, a cephalic presentation was achieved after ECV in 48 (53.3%). In multiparous women, ECV was successful in 32 (78.0%).

### 3.1. Delivery Outcomes

Data of delivery were not available for three pregnant women who gave birth in other hospitals. After ECV, the delivery occurred at 38.5 weeks gestation (SD 0.5) on average (Table 2).

An elective scheduled cesarean section due to non-cephalic presentation occurred in 46 pregnant women (35.9%). An urgent cesarean section during the first 24 h after ECV was required in 15 (11.5%) women.

Finally, after a successful ECV, a spontaneous delivery was achieved in 38 women (56.7%). Eighteen pregnant women (26.9%) required an operative delivery, and an intrapartum urgent cesarean section was performed in 11 women (16.4%).

### 3.2. Complications

A major complication arose in 15 pregnant women (11.5%) (Table 3). Two newborns were admitted to neonatal care unit, and one was admitted to neonatal ICU. Neither of these three cases was related directly to ECV.

One pregnancy had an operative delivery at 38 + 1 weeks of gestation, two weeks after ECV. During the labor, fetal bradycardia required an operative delivery. The newborn APGAR score at the first and fifth minutes of life were 5/9, the fetal cord pH was 7.11 (arterial) and 7.21 (venous). The newborn was admitted to neonatal unit care. In the second case, an urgent cesarean section was performed for a non-reassuring fetal heart rate pattern at 37 + 6 weeks of gestation (5 days after ECV). The newborn APGAR score was 3/5, the fetal cord as 7.08 (arterial) and 7.12 (venous). The newborn was admitted to the neonatal ICU. The last case was a pregnancy with a spontaneous delivery at 40 + 3 weeks of gestation (3 weeks after ECV). A shoulder dystocia occurred during the delivery. The newborn weighed 4120 g, the APGAR score was 5/8, fetal cord pH was 7.05 (arterial) and 7.12 (venous). The newborn was admitted to neonatal unit care.

One woman suffered bronchoaspiration. The bronchoaspiration occurred just after ending the ECV. The woman was admitted to the maternal unit care with antibiotic treatment. Although a cephalic presentation was achieved; finally, a cesarean section was performed due to the bronchoaspiration after seven days with treatment. A female was born with APGAR 9/10, vein cord pH = 7.32. The woman and her newborn were discharged with no sequelae.

## 4. Discussion

In this study, ECV was performed with tocolysis with ritodrine, sedation with propofol and analgesia with paracetamol. This is the first published study in which propofol is considered as a sedative agent in ECV.

ECV is a safe and effective procedure for achieving a cephalic presentation. The ECV success rate in this study was 61.1%. The success rate reported was higher than those published without analgesia (49.0%) [3], with systemic opioids (51.7%) [17], and similar to epidural analgesia (58.4%) [12]. Burgos et al. [17] advised that although remifentanil did not increase the ECV success rate, the pain related to the procedure was markedly reduced. To the best of our knowledge, no report using propofol has been published. Magro-Malosso et al. [12] found that spinal or epidural analgesia increased the ECV success rate (RR 1.54 95%CI 1.22–2.63) compared with systemic opioids. Probably, sedation with propofol may achieve an adequate maternal unconsciousness that facilitate the maneuver.

In nulliparous women, cephalic presentation in this study was achieved after ECV in 51.5%, which is a higher rate than those found in the bibliography (40%) [3]. Equally, in multiparous women, ECV in this study was successful in 74.0%, and it was higher than those found in other reports (64%) [3]. This difference may be due to the use of propofol, the use of ritodrine as a tocolytic agent just before the procedure, the gestational age at which ECV was performed, and the obstetrician experience.

Propofol is a non-teratogenic [18] anesthetic agent appropriated for short-duration surgical procedures [19]. Maternal hypotension is a common adverse effect of propofol [20]. In our study, although clinically relevant hypotension occurred in 18.3%, any non-reassuring fetal heart rate pattern was registered. Propofol was supposed to raise neonatal sedation and depression [21], but no differences were reported in APGAR score and neurological and adaptative capacity scores when propofol (2 mg/kg) and other techniques, such as spinal anesthesia or barbiturates, were compared [22,23,24]. In our study, the propofol mean dose was 156.1 mg, which approximately corresponded with 1.5–2.0 mg/kg.

Other studies, that have performed spinal techniques for analgesia, have reported a lower ECV success rate [10,11,12]. However, two recent studies should be noted since the ECV success rate when neuraxial anesthesia was performed was similar to this [13,25,26]. Although Weiniger et al. [25] reported an ECV success rate of 87.1% in pregnant women receiving spinal analgesia, they excluded women with a previous cesarean section and women with a BMI above 40 kg/m^2^. Khaw et al. [26] reported an ECV success rate of 52.0% in pregnant women receiving spinal analgesia and 40% in pregnant women receiving intravenous remifentanil.

No more than two attempts were proposed to perform ECV in this protocol. The National Society of Gynecology and Obstetrics recommends [27] no more than four attempts with the objective to avoid abruptio placentae and fetal heart rate disturbance [28]. Since the maneuver is facilitated by tocolysis and sedation, obstetricians might be induced to apply smaller forces. Because of that, attempts in this study were limited to two in order to be more cautious with the procedure.

The major ECV complications rate in this study was 11.5%. ECV complications arose more than without analgesia (5.39%) [3]. Burgos et al. [17] reported a similar ECV complications rate (20.3%) when systemic opioids were used. Magro-Malosso et al. [12] found fetal heart rate disturbances (transient bradycardia and other non-reassuring fetal heart rate patterns) in 18.7%. Minor complications such as premature rupture of membranes, minor vaginal bleeding, or symptomatic uterine contractions during the following 48 h, were not considered in other reports. Non-reassuring fetal heart rate patterns occurred in 4.7% of the procedures; this rate is higher than without analgesia (0.9%) [3] but similar or lower than systemic opioids or spinal analgesia (around 20%) [12,17]. An urgent cesarean section rate during the first 24 h after ECV in this study was needed in 11.5% of the cases. A lower urgent cesarean section during the first 24 h after ECV (1.6%) was reported when spinal anesthesia or systemic opioids were used [12,17]. These differences may be explained by tocolysis and sedation. Although both of them help the obstetricians to apply smaller forces on the maternal abdomen, they can also induce maternal hypotension with clinical consequences (CTG abnormalities or vaginal bleeding).

Even though a maternal bronchoaspiration was reported, it should be highlighted that this was the only severe adverse event that occurred during the seven years for which we have ECV records (more than 690 procedures). Notwithstanding that fasting is recommended for at least eight hours before the procedure, delayed gastric emptying can occur during pregnancy [29]. This case enhances the need for a toughly pre-anesthesia consult.

Some strengths should be highlighted. This study recruited a large number of pregnant women. All the ECV procedures were performed by the same obstetricians, anesthesiologists, and midwives that constitute the ECV working group, with more than seven years of experience in ECV. All the procedures were carried out in the operating room where an urgent cesarean section can be rapidly performed. All of the professionals who composed the ECV working group were trained to manage any complication that could arise and have regularly continued training.

This research, however, is subject to several limitations. An important limitation of our study is the nature of the study design. These results need to be confirmed in further experiments. Another relevant limitation was the loss of information about anesthetic procedure. Due to logistical issues, some data for anesthesia techniques (drug dose) were unavailable for 89 women who were excluded from the analysis. No complications arose in any of them, but the drug dose and blood pressure register were not available for analysis for those women. Probably, if those patients had been included (*n* = 220), the complications rate and cesarean section during 24 h after the procedure would have decreased substantially (6.7% and 12.8%, respectively).

Another limitation of our study is that the maternal weight was measured at 12 weeks gestation when the first trimester scan was performed. The weight modifications during pregnancy were not taken into consideration.

## 5. Conclusions

The results of this study showed that the administration of propofol, facilitates ECV. Studies comparing the effectiveness and safety of the different anesthetic adjunct, for ECV, are urgently required.

## Figures and Tables

**Table 1 jcm-11-00489-t001:** Characteristics and obstetrics outcomes of the pregnant women who underwent external cephalic version (ECV).

Variable	Mean/Frequency, %	±SD/Count
Maternal Age, years	32.1	±0.4
Gestational Age at ECV, weeks	37.3	±0.04
Nulliparity	68.7	90
Previous Cesarean Section	3.1	4
Maternal BMI, Kg/m^2^	27.5	±0.4
BMI < 25, Kg/m^2^	30.8	36
BMI 25–30, Kg/m^2^	40.2	47
BMI 30–35, Kg/m^2^	21.4	25
BMI > 35, Kg/m^2^	7.7	9
Estimated Fetal Weight before ECV, g	2754.4	±29
Placental position		
Anterior	56.6	73
Posterior	34.9	45
Fundus	3.1	4
Lateral Wall	5.4	7
Amniotic Fluid Pocket, mm	50.3	±1.4
Fetal position		
Breech	95.4	125
Transverse lie	4.6	6
ECV Success	61.1	80

**Table 2 jcm-11-00489-t002:** Delivery outcomes after ECV.

Variable	Mean/Frequency, %	±SD/Count
Gestational Age at Birth, weeks	38.5	±0.5
Spontaneous Delivery	56.7 ^1^	38
Operative Delivery	26.9 ^1^	18
Urgent Cesarean Section	16.4 ^1^	11
Elective Cesarean Section	35.9	46
Cesarean Section during the following 24 h of ECV	11.5	15
Newborn Weight, g	3251.5	±37.7
APGAR Score at 1’ < 7	2.5	2
APGAR Score at 5’ < 7	1.3	1
ECV Major Complications	11.5	15

^1^ Relative frequency of pregnant women with successful ECV.

**Table 3 jcm-11-00489-t003:** ECV complications. FHR: fetal heart rate.

Variable	Frequency, %	Count
Non-Reassuring FHR Pattern	4.6	6
Major Vaginal Bleeding	4.6	6
Minor Vaginal Bleeding	3.8	5
Uterine Contractions	2.3	3
Cord Prolapse	1.5	2
Premature Rupture of Membranes	0.8	1
Maternal Bronchoaspiration	0.8	1

## Data Availability

The data presented in this study are available on request from the corresponding author. The data are not publicly available due to privacy concerns.

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
