# Peer review of "Sedation with Propofol plus Paracetamol in External Cephalic Version: An Observational Study"

_jcm, 2022, doi:10.3390/jcm11030489_

Round 1

Reviewer 1 Report

The authors conducted a prospective observational pilot study and recruited 131 (out of 242) parturients in two years. Propofol (156.1 mg) was used for those parturients receiving ECV maneuver. Several study parameters were acquired and analyzed, including risks and factors of, successful rate of ECV, conversion rate to C-section, obstetrical outcomes, maternal and fetal outcomes, etc. The authors therefore concluded as “Deep sedation with propofol could be an adequate option for ECV. ECV outcomes when propofol was used were similar to those with other sedative agents. More studies are needed to compare its effectiveness and safety with neuraxial techniques.”

Reviewer’s comments:

Major comment:

The maneuver of ECV may be impeded by uterine contraction and abdominal muscle contraction. Both contractions then cause severe pain and need adequate analgesia to relieve. Therefore, to obtain high success rate of ECV needs both effective tocolytics and ANALGESIA (e.g., regional analgesia or anesthesia, systemic use of opioids, inhalational anesthesia etc). On the other hand, some practices did not use any anesthesia or analgesia and still gained 60-80% of success rate of version. The intention of the authors was to apply “deep sedation”, on top of paracetamol (1 gm), to overcome the pain intensity caused by ECV. Therefore, the author need to address the following issues:

(a)    Whether this study purpose should be to explore “both the analgesic effect of paracetamol plus the sedative effect of propofol on ECV”?

(b)    Regarding “deep sedation”, the authors need to explain how to define the level of “deep sedation” in this study, instead of simply mentioned 156 mg propofol was given to the patients? Any BIS or other neuromonitoring indices could be provided? Any correlation between the relationship between administering 156 mg propofol and estimated effect-site concentration of propofol in PK simulation model or other physical signs?

(c)    Although there is only one case (out of 131 ECV cases) of aspiration pneumonia, the incidence is not low and should not be under-stated.

(d)    In the cited references (9-13, 17, 25-26), none of them are related to the effects of “sedatives” for ECV. In contrast, all the interventions of analgesia and anesthesia for ECV (spinal anesthesia, spinal analgesia, epidural analgesia, combined spinal/epidural analgesia, systemic remifentanil and other opioids, inhalational anesthetics, etc) in the literature are not related to sedation effect. Therefore, it might be better for the authors to explain more why propofol itself can compete with those modalities for ECV.

(e)    Following argument in (d), the authors should clarify the statement in their conclusion that “ECV outcomes when propofol was used were similar to those with other sedative agents.” is supported by which references.

(f)    The authors stated that propofol 2 mg/kg (150 mg) is “low dose”. For example, “propofol 0.8 mg/kg followed by a continuous infusion of propofol 30 microg/kg/min” is regarded as “low dose”. For procedural sedation, 50-100 ug/kg/min following 0.3 mg/kg bolus of propofol could also be regarded as low dose.

Author Response

RESPONSE TO REVIEWER 1 COMMENTS

Ms. Ref. No.: Birth-20-05-41

Manuscript ID: jcm-1542183

Title: Deep Sedation with Propofol in External Cephalic Version: A Pilot Study

Major comment:

The maneuver of ECV may be impeded by uterine contraction and abdominal muscle contraction. Both contractions then cause severe pain and need adequate analgesia to relieve. Therefore, to obtain high success rate of ECV needs both effective tocolytics and ANALGESIA (e.g., regional analgesia or anesthesia, systemic use of opioids, inhalational anesthesia etc). On the other hand, some practices did not use any anesthesia or analgesia and still gained 60-80% of success rate of version. The intention of the authors was to apply “deep sedation”, on top of paracetamol (1 gm), to overcome the pain intensity caused by ECV. Therefore, the author need to address the following issues:

Point 1: Whether this study purpose should be to explore “both the analgesic effect of paracetamol plus the sedative effect of propofol on ECV”?

Response 1: Thank you for your comment and suggestion. We appreciate your suggestions. In our study, propofol was used as sedative agent in combination with an analgesic drug (paracetamol). It was detailed in Methods paragraph. We opted to omit the “analgesic effect of paracetamol” in the objectives because we preferred to highlight the novelty of using propofol in ECV. However, it can be disappointing to omit it and we have modified the title adding the analgesic effect of paracetamol.

Point 2: Regarding “deep sedation”, the authors need to explain how to define the level of “deep sedation” in this study, instead of simply mentioned 156 mg propofol was given to the patients? Any BIS or other neuromonitoring indices could be provided? Any correlation between the relationship between administering 156 mg propofol and estimated effect-site concentration of propofol in PK simulation model or other physical signs?

Response 2: Thank you for your comment and suggestion. We appreciate your suggestions. During the procedure, there was continuous feedback between obstetric and anesthesiologist teams. With the objective of achieving an unconsciousness state, an initial dose of 50 mg of propofol was administered in bolus at the beginning of the procedure. Then during the maneuver, if a deeper sedation was required (excessive maternal responsiveness to stimulation), additional boluses of 10 to 20 mg propofol were administered at intervals of at least 20 to 30 seconds.

During the maneuver, a clinical evaluation of consciousness level was performed. Thanks to the continuous feedback between obstetric and anesthesiologist teams and to the brief procedure, we opted to not use Bispectral Index (BIS). Because of that, we opted to omit “deep” sedation in order to clarify the used terms.

We did not carry out any intravenous drug monitoring in this study either any simulation model was considered before the study. We reported a mean dose of 156.1 mg of propofol to correctly achieve sedation. The minimum dose administered was 50 mg. The maximum dose administered was 350 mg. Finally, the median was 150 mg. We did not find any statistically significant relationship between propofol dose and clinically relevant hypotension either requirement of vasoactive drugs or nausea.

Point 3: Although there is only one case (out of 131 ECV cases) of aspiration pneumonia, the incidence is not low and should not be under-stated.

Response 3: Thank you for your comment and suggestion. We appreciate your suggestions. We were extremely worried about this patient. Although she was fasted, a bronchoaspiration occurred. To be transparent and to show the potential risks of sedation, we reported in detailed this complication. However, to be honest, we have to contextualize the complication frequency. The ECV super-specialized team was created in 2014. From this year to nowadays more than 700 maneuvers were carried out, and this is the only bronchoaspiration suffered. Furthermore, we recommended to enhance pre-anesthesia consult for all the obstetrics procedures.

Point 4: In the cited references (9-13, 17, 25-26), none of them are related to the effects of “sedatives” for ECV. In contrast, all the interventions of analgesia and anesthesia for ECV (spinal anesthesia, spinal analgesia, epidural analgesia, combined spinal/epidural analgesia, systemic remifentanil and other opioids, inhalational anesthetics, etc) in the literature are not related to sedation effect. Therefore, it might be better for the authors to explain more why propofol itself can compete with those modalities for ECV.

Response 4: Thank you for your comment and suggestion. We appreciate your suggestions. We apologize for not contextualizing enough the chances that propofol may offer to facilitate ECV. We would like to emphasize that propofol, as a sedative agent, may achieve an adequate maternal unconscious state and it may facilitate the maneuver enough and, subsequently smaller forces were applied at the maternal abdomen. We added an explanation in the discussion paragraph.

All the studies previously published used neuraxial techniques, systemic opioids or inhalational anesthetics. The main goal of these techniques (analgesia and tocolysis) is to avoid maternal opposition to the manual breech rotation. We considered that sedation is a good alternative for reaching maternal unconsciousness.

Point 5: Following argument in (d), the authors should clarify the statement in their conclusion that “ECV outcomes when propofol was used were similar to those with other sedative agents.” is supported by which references.

Response 5: Thank you for your comment and suggestion. We appreciate your suggestions. We also answered above. In this case (conclusions), we opted to modify this sentence.

Point 6: The authors stated that propofol 2 mg/kg (150 mg) is “low dose”. For example, “propofol 0.8 mg/kg followed by a continuous infusion of propofol 30 microg/kg/min” is regarded as “low dose”. For procedural sedation, 50-100 ug/kg/min following 0.3 mg/kg bolus of propofol could also be regarded as low dose.

Response 6: Thank you for your comment and suggestion. We appreciate your suggestions. The authors [22-24] defined their propofol dose as low dose. We opted to omit “low dose” in the discussion paragraph in order to clarify the used terms.

Reviewer 2 Report

  1. After reading the whole article, this is not a pilot study. It is just an observational study. Please see BMJ 2016;355:i5239 for the definition of a pilot study.
  2. In the abstract: when reporting the percentage, please also give numerator and denominator.
  3. Page 3/8, lines 113-115: “This study was approved 30 April 2020 by the Clinical Research Committee of the 'Virgen de la Arrixaca' University Clinical Hospital (2020-5-6-HCUVA). Written informed consent was obtained from all participants.” I suggest that this sentence be moved to the first paragraph of the Methods section.
  4. I note that 242 pregnant women underwent ECV during the study period. Is it possible that the authors compare the outcomes between women who were given propofol sedation and those who were not?
  5. Language revision required. For example: “… several studies have shown similar APGAR scores and neurological and adaptive capacity scores with the use of low dose (2 mg/kg) propofol and with spinal anesthesia or barbiturates (Page 5/8, lines 192-194)”, “Other reports did not report minor complications… (page 6/8, line 214)”, and “These differences may be since tocolysis and sedation might induce the obstetricians to apply greater forces or hypotension caused by anesthesia techniques that might cause CTG abnormalities or vaginal bleeding (page 6/8, lines 221-224)”. Consider revision to make the expression clearer.
  6. Page 5/8, line 199: “Although C. Weiniger et al. [25] reported …” Only family name is required here and in other places when reporting author’s name.

Author Response

RESPONSE TO REVIEWER 2 COMMENTS

Ms. Ref. No.: Birth-20-05-41

Manuscript ID: jcm-1542183

Title: Deep Sedation with Propofol in External Cephalic Version: A Pilot Study

Point 1: After reading the whole article, this is not a pilot study. It is just an observational study. Please see BMJ 2016;355:i5239 for the definition of a pilot study.

Response 1: Thank you for your comment and suggestion. We apologize for not explaining it in deep detail previously. After reading the “CONSORT 2010 statement”, we opted to change the study definition. Initially, we considered this study as a pilot since it was the first time that propofol was used for ECV. But, to be honest, in our manuscript pilot was a misleading definition.

Point 2: In the abstract: when reporting the percentage, please also give numerator and denominator.

Response 2: Thank you for your comment and suggestion. We appreciate your suggestions. Although we opted to describe qualitative variables with count and relative frequency, the way you proposed seems to be more representative. We have added every proportion in the abstract.

Point 3: Page 3/8, lines 113-115: “This study was approved 30 April 2020 by the Clinical Research Committee of the 'Virgen de la Arrixaca' University Clinical Hospital (2020-5-6-HCUVA). Written informed consent was obtained from all participants.” I suggest that this sentence be moved to the first paragraph of the Methods section.

Response 3: Thank you for your comment and suggestion. We appreciate your suggestions. We also agree with you that this sentence would fit better in the first paragraph. We have modified it.

Point 4: I note that 242 pregnant women underwent ECV during the study period. Is it possible that the authors compare the outcomes between women who were given propofol sedation and those who were not?

Response 4: Thank you for your comment and suggestion. We appreciate your suggestions. Unfortunately, we have not yet performed this study. In the short-term future we would like to design a clinical trial comparing propofol with other techniques (such as neuraxial techniques). However, we considered that firstly we have to accomplish an observational study just with propofol.

Point 5: Language revision required. For example: “… several studies have shown similar APGAR scores and neurological and adaptive capacity scores with the use of low dose (2 mg/kg) propofol and with spinal anesthesia or barbiturates (Page 5/8, lines 192-194)”, “Other reports did not report minor complications… (page 6/8, line 214)”, and “These differences may be since tocolysis and sedation might induce the obstetricians to apply greater forces or hypotension caused by anesthesia techniques that might cause CTG abnormalities or vaginal bleeding (page 6/8, lines 221-224)”. Consider revision to make the expression clearer.

Response 5: Thank you for your comment and suggestion. We appreciate your suggestions. We opted to use one of the editing services offered by mdpi. We attached the English editing service certificate. Although we are not native English speakers, it is also evident that some language issues can be clearer. We have modified those you mentioned and others we have also detected.

Point 6: Page 5/8, line 199: “Although C. Weiniger et al. [25] reported …” Only family name is required here and in other places when reporting author’s name.

Response 6: Thank you for your comment and suggestion. We appreciate your suggestions. We strongly apologized for this mistake. We have immediately modified it.

Round 2

Reviewer 1 Report

Well done for the revision. Congratulations!

Hope your story will eventually benefit the pregnant women who need ECV maneuver during the course.

Author Response

Well done for the revision. Congratulations!

Hope your story will eventually benefit the pregnant women who need ECV maneuver during the course.

We would like to be grateful for this chance for improvement. We hope we will carry out the clinical trial comparing propofol with neuraxial techniques as soon as posible.

Reviewer 2 Report

  1. Abstract: “External Cephalic Version”: Please use lower case letter. This is not a proper noun.
  2. Page 2/line 49: in the cited references, the control groups were administered with opioids. Opioid analgesia should not be described as sedation.
  3. Page 5/line 198-200: “This difference may be due to the use of propofol, the use of ritodrine, as a tocolytic agent, just before the procedure, the gestational age at which ECV was performed, or the obstetrician experience.” Consider to revise the sentence as: “This difference may be due to the use of propofol, the use of ritodrine as a tocolytic agent just before the procedure, the gestational age at which ECV was performed, and the obstetrician experience.”
  4. An important limitation is the nature of the study design, i.e., an observational study.

Author Response

RESPONSE TO REVIEWER 2 COMMENTS

Manuscript ID: jcm-1542183

Title: Deep Sedation with Propofol in External Cephalic Version: A Pilot Study

Point 1: Abstract: “External Cephalic Version”: Please use lower case letter. This is not a proper noun.

Response 1: Thank you for your comment and suggestion. We apologize for this issue. We immediately have changed it.

Point 2: Page 2/line 49: in the cited references, the control groups were administered with opioids. Opioid analgesia should not be described as sedation.

Response 2: Thank you for your comment and suggestion. We appreciate your suggestions. We apologize for this issue. It was an evident mistake we committed during the last review. All the studies reported have used neuraxial techniques or systemic opioids (fentanyl or remifentanil). We have immediately changed it.

Point 3: Page 5/line 198-200: “This difference may be due to the use of propofol, the use of ritodrine, as a tocolytic agent, just before the procedure, the gestational age at which ECV was performed, or the obstetrician experience.” Consider to revise the sentence as: “This difference may be due to the use of propofol, the use of ritodrine as a tocolytic agent just before the procedure, the gestational age at which ECV was performed, and the obstetrician experience.”

Response 3: Thank you for your comment and suggestion. We appreciate your suggestions. We strongly apologize for such misleading grammar use. We also agree this sentence could be clarified. We have modified it.

Point 4: An important limitation is the nature of the study design, i.e., an observational study.

Response 4: Thank you for your comment and suggestion. We appreciate your suggestions. This is an observational study with its inherent limitations that must be also highlighted. We have modified limitations paragraph.